# Canonical and non-canonical PRC1 differentially contribute to regulation of neural stem cell fate

Janine Hoffmann[1], Theresa M Schütze[1], Annika Kolodziejczyk[1], Karolin Küster[1], Annekathrin Kränkel[2],
Susanne Reinhardt[2], Razvan P Derihaci[3,4], Cahit Birdir[3,5], Pauline Wimberger[3,4], Haruhiko Koseki[6], Mareike Albert[1]

**Neocortex development is characterized by sequential phases of neural progenitor cell (NPC) expansion, neurogenesis, and gliogenesis. Polycomb-mediated epigenetic mechanisms are known to play important roles in regulating the lineage potential of NPCs during development. The composition of Polycomb repressive complex 1 (PRC1) is highly diverse in mammals and was hypothesized to contribute to context-specific regulation of cell fate. Here, we have performed a side-by-side comparison of the role of canonical PRC1.2/1.4 and non-canonical PRC1.3/1.5, all of which are expressed in the developing neocortex, in NSC proliferation and differentiation. We found that the deletion of *Pcgf2/4* in NSCs led to a strong reduction in proliferation and to altered lineage fate, both during the neurogenic and gliogenic phase, whereas *Pcgf3/5* played a minor role. Mechanistically, genes encoding stem cell and neurogenic factors were bound by PRC1 and differentially expressed upon *Pcgf2/4* deletion. Thus, rather than different PRC1 subcomplexes contributing to different phases of neural development, we found that canonical PRC1 played a more significant role in NSC regulation during proliferative, neurogenic, and gliogenic phases compared with non-canonical PRC1.**

## Introduction

During the development of the neocortex, stem and progenitor cells initially proliferate, before sequentially giving rise to neurons destined to different cortical layers, and subsequently generate astrocytes and oligodendrocytes (Qian et al, 2000; Lodato & Arlotta, 2015). Precise spatial and temporal regulation of NPC proliferation and differentiation is key for the proper formation of the intricate structure of the neocortex. Transcription factors and epigenetic mechanisms play important roles in orchestrating dynamic changes in gene expression that underlie coordinated neural differentiation programs in the developing neocortex (Albert & Huttner, 2018; Tsuboi et al, 2019; Desai & Pethe, 2020).

Chromatin modifiers of the Trithorax and Polycomb groups maintain active and repressed gene activity states during embryonic development (Ringrose & Paro, 2004; Piunti & Shilatifard, 2016; Schuettengruber et al, 2017). Polycomb proteins assemble into two major complexes, PRC1 and PRC2, which catalyse mono-ubiquitination of histone 2A lysine 119 (H2AK119ub1) and trimethylation of histone 3 lysine 27 (H3K27me3), respectively. These complexes are important determinants of the ability of NPCs to either proliferate or to give rise to neurons or glial cells (Hirabayashi et al, 2009; Tyssowski et al, 2014), and mutations in Polycomb components were reported to cause neurodevelopmental disorders (Mastrototaro et al, 2017; Pierce et al, 2018; Bölicke & Albert, 2022).

Specific deletion of the PRC2 histone methyltransferase *Ezh2* in the early developing neocortex causes an up-regulation of gene expression and a shift of apical radial glial fate from self-renewal to differentiation (Pereira et al, 2010), reducing the neuronal output and leading to a substantially smaller neocortex. Moreover, deletion of *Ring1b*, an integral component of PRC1, during the neurogenic phase results in altered neuronal subtype specification (Morimoto-Suzki et al, 2014). In this context, the E3 ubiquitin ligase activity of Ring1b was found to be necessary for the temporary repression of key neuronal genes in neurogenic NPCs (Tsuboi et al, 2018). These data indicate that Polycomb complexes control important aspects of corticogenesis.

Epigenome profiling in specific neural cell populations isolated from the mouse developing neocortex has revealed dynamic changes in H3K4me3 and H3K27me3 during neocortical lineage specification (Albert et al, 2017). An important question is how Polycomb target gene specificity is achieved in different neocortical cell types. One way to dynamically control Polycomb function and targeting is by altering the composition of Polycomb complexes, which in mammals is highly diverse, enabling the assembly of various subcomplexes with different functionalities (Luis et al, 2012;

[1]Center for Regenerative Therapies Dresden, TUD Dresden University of Technology, Dresden, Germany   [2]DRESDEN-Concept Genome Center, Center for Molecular and Cellular Bioengineering, Technology Platform of the TUD Dresden University of Technology, Dresden, Germany   [3]Department of Gynecology and Obstetrics, Technische Universität Dresden, Dresden, Germany   [4]National Center for Tumor Diseases, Dresden, Germany   [5]Center for Feto/Neonatal Health, Technische Universität Dresden, Dresden, Germany   [6]Laboratory of Developmental Genetics, RIKEN Center for Integrative Medical Sciences, Yokohama, Japan

Correspondence: mareike.albert@tu-dresden.de

Tsuboi et al, 2019; Kim & Kingston, 2020). During neocortex development, chromatin remodellers of the chromodomain helicase DNA–binding (Chd) family, which interact with PRC2 complexes, show differential expression. Whereas Chd5 is expressed in neurons and controls neuronal differentiation (Egan et al, 2013), Chd4 is expressed in neural progenitor cells during early neurogenesis, where it functions in the inhibition of astroglial differentiation (Sparmann et al, 2013). Such a switch in subunit composition may contribute to the retargeting of PRC2 during neocortex development.

PRC1 complexes are classified as canonical or non-canonical, depending on which of the complex members are included, with all complexes containing the central Ring1a/b E3 ubiquitin ligase core (Piunti & Shilatifard, 2016; Tsuboi et al, 2019; Blackledge & Klose, 2021). Canonical PRC1 complexes (PRC1.2/1.4) contain Pcgf2/4 (also known as Mel-18/Bmi1), one of three polyhomeotic (Phc) proteins and one of five chromobox (Cbx) proteins that recognize H3K27me3 mediated by PRC2. Non-canonical PRC1 is targeted to chromatin independently of H3K27me3 and is characterized by the inclusion of Pcgf1 (PRC1.1), Pcgf3/5 (PRC1.3/1.5), or Pcgf6 (PRC1.6) (Blackledge & Klose, 2021), even though non-canonical PRC1.2/1.4 lacking Cbx and Phc has also been described (Gao et al, 2012). Non-canonical PRC1 has high ubiquitin ligase activity, whereas canonical PRC1 was reported to promote higher order chromatin structures, but to display low ligase activity and low contribution to target gene repression (Fursova et al, 2019; Blackledge & Klose, 2021).

In embryonic stem cells, the interchange of Cbx family proteins in PRC1 (Kim & Kingston, 2020) has been reported to modulate the balance between self-renewal and lineage commitment (Morey et al, 2012; O'Loghlen et al, 2012; Santanach et al, 2017), and different Cbx paralogs are required for different cell lineages (Luis et al, 2011; Klauke et al, 2013). Likewise, Pcgf homologs were suggested to promote context- and stage-specific functions during differentiation and development (Morey et al, 2015; Kloet et al, 2016).

Although the canonical PRC1 components Pcgf2 and Pcgf4 (Akasaka et al, 2001; Molofsky et al, 2003, 2005; Leung et al, 2004; Zencak et al, 2005; He et al, 2009), as well as the non-canonical PRC1 components Pcgf3 and Pcgf5 (Gao et al, 2014; Yao et al, 2018; Meng et al, 2020), have been implicated in neural differentiation and brain development, here we set out to perform a systemic comparative analysis of the role of different PRC1 subcomplexes in NSC proliferation and differentiation. Specifically, we deleted canonical (*Pcgf2/4*) and non-canonical PRC1 (*Pcgf3/5*) in proliferating, neurogenic, and gliogenic NSCs to elucidate the function of different PRC1 subcomplexes in key phases of cortical development.

# Results

## Pcgf homologs are differentially expressed in the mouse and human developing neocortex

To analyse the expression of canonical and non-canonical PRC1 components (Fig 1A) in the human developing neocortex (Fig 1B), we first mined RNA-seq data of microdissected human foetal cortex (Fietz et al, 2012). The core components of PRC1, *RING1A* and *RING1B*, showed a fairly uniform expression across the germinal zones (VZ, ISVZ, OSVZ), which are enriched in NPCs, and the cortical plate, where neurons reside (Fig 1C). This was confirmed by immunohistochemistry of the human foetal tissue, in which RING1B and H2AK119ub1 showed a comparable expression across all cortical layers with a minor increase in the ventricular zone (VZ) and cortical plate (CP) (Fig 1D and E). In contrast, *PCGF2*, *PCGF3*, and *PCGF4* were expressed at higher levels in the CP compared with germinal zones, whereas *PCGF5* was specifically enriched in the VZ (Fig 1C–E). *PCGF1* and *PCGF6* showed no or low expression in the VZ (Fig S1A–D). These data indicate that PCGF homologs are differentially expressed across neural cell types of the human developing neocortex.

Next, we analysed the expression of PRC1 components in the mouse developing neocortex (Fig 1F). Ring1a/b and H2AK119ub1 were uniformly distributed across all zones, with slight enrichment in the VZ and CP (Fig 1G–I). In analogy to the human developing neocortex, Pcgf2, *Pcgf3*, and Pcgf4 showed some enrichment in the CP, whereas *Pcgf5* was specifically expressed in the VZ (Fig 1G–I), even though the differences were less pronounced in the mouse compared with the human developing neocortex. The results are in line with previous studies noting the high abundance of Pcgf2 and Pcgf4 in NPCs (Tagawa et al, 1990; Zencak et al, 2005), whereas the neuronal expression of Pcgf4 was not seen previously (Zencak et al, 2005).

Taken together, the canonical PRC1 components Pcgf2/4 were enriched in neurons compared with NPCs, which is interesting giving their previous implication in the regulation of NSC self-renewal and proliferation (Molofsky et al, 2003, 2005; Zencak et al, 2005; He et al, 2009). The non-canonical PRC1 components Pcgf3 and Pcgf5 showed differential enrichment in neurons versus NPCs, respectively.

## Canonical and non-canonical PRC1 contribute to different degrees to the regulation of NSC proliferation

To systematically compare the role of canonical and non-canonical PRC1 in NSC proliferation, we isolated NPCs from the developing dorsolateral neocortex of E12.5 mouse embryos from either *Pcgf2*$^{F/F}$; *Pcgf4*$^{F/F}$; *Nes*::CreERT2/+ or *Pcgf3*$^{F/F}$; *Pcgf5*$^{F/F}$; *Nes*::CreERT2/+ mouse lines (Imayoshi et al, 2006; Almeida et al, 2017; Fursova et al, 2019) and induced the deletion of *Pcgf* genes in vitro by the addition of 4-hydroxytamoxifen (OHT) (Fig 2A). Deletion of *Pcgf* genes was highly efficient (Fig 2B and C) and resulted in a complete loss of Pcgf3, Pcgf4, and Pcgf5 proteins after 3 d in vitro (DIV), and a reduction in Pcgf2 levels (Fig 2D and E). Deletion of *Pcgf2/4* resulted in reduced global Ring1b levels without changes in H2AK119ub1 (Fig 2F and G), whereas deletion of *Pcgf3/5* did not affect Ring1b levels but resulted in reduced H2AK119ub1 (Fig 2H and I). These results mirror previous findings in mouse embryonic stem cells (Fursova et al, 2019) and are consistent with weak ligase activity of canonical PRC1 (Gao et al, 2012; Blackledge et al, 2014).

After the validation of conditional knockout (cKO) in NSCs, we then asked how deletion of canonical (*Pcgf2/4*) and non-canonical (*Pcgf3/5*) PRC1 affects NSC proliferation. Whereas deletion of *Pcgf2/4* led to an almost complete inability of NSCs to proliferate (Fig 2J–L), the deletion of *Pcgf3/5* had a more modest effect (Fig 2M and N). Overall, the cell numbers were reduced to less than 10% of control

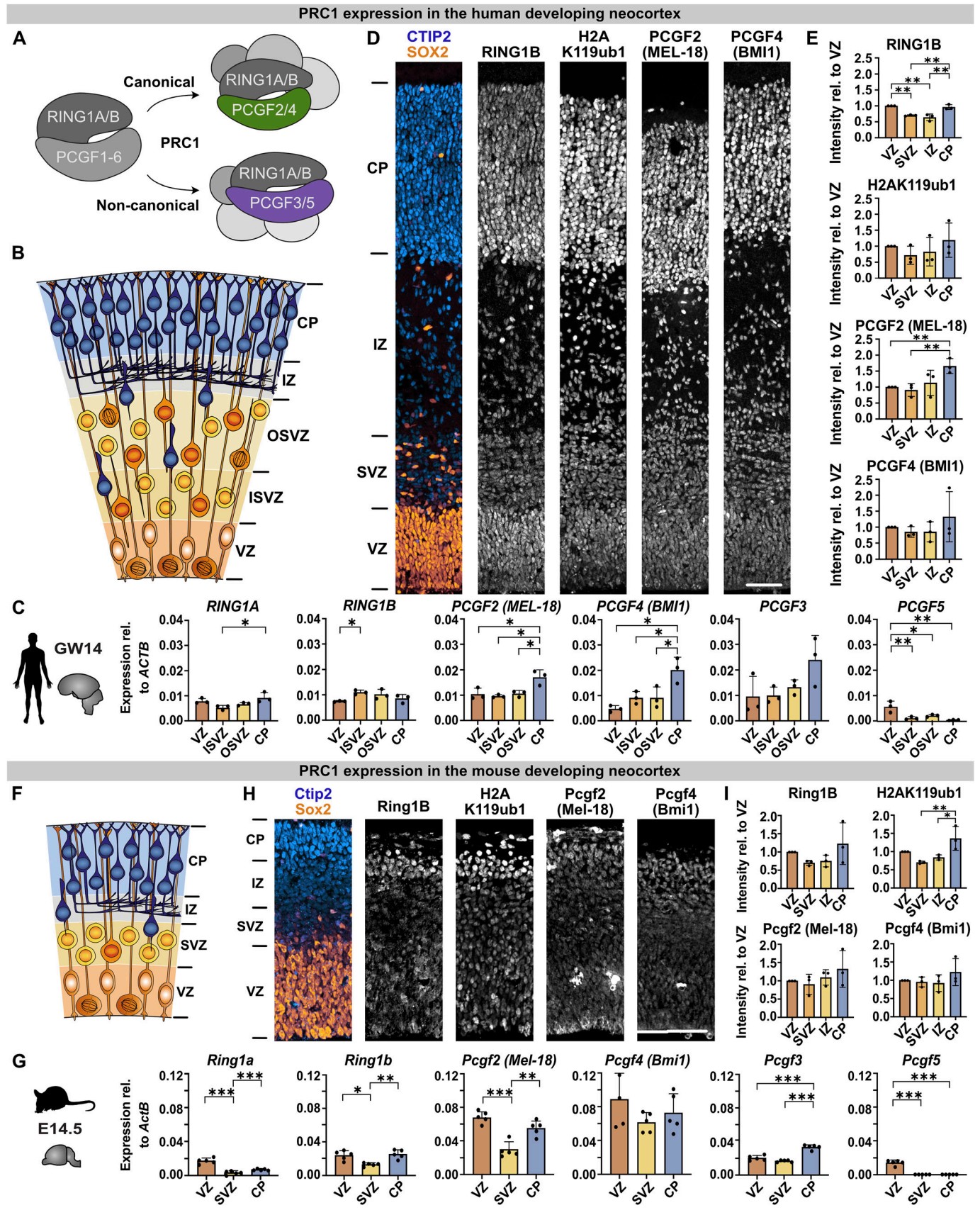

after 6 DIV for *Pcgf2/4* cKO compared with roughly 70% of control for *Pcgf3/5* cKO (Fig 2O), highlighting the differential contribution of canonical and non-canonical PRC1 to the regulation of NSC proliferation.

These results are in agreement with previous reports on the role of *Pcgf4* in regulating NSC self-renewal and proliferation (Molofsky et al, 2003, 2005; Zencak et al, 2005; He et al, 2009). Pcgf2 was reported to function antagonistically to Pcgf4 by promoting cell senescence through down-regulation of Pcgf4 (Guo et al, 2007). Yet, double knockout of *Pcgf2/4* revealed the synergistic effect of both genes, resulting in strongly exacerbated phenotypes compared with single mutant mice (Akasaka et al, 2001). In line with this, deletion of *Pcgf2/4* in NSCs resulted in an almost complete loss of the ability of NSCs to proliferate. Moreover, *Pcgf3* has previously been implicated in the regulation of tumour cell proliferation (Hu et al, 2021), whereas *Pcgf5* and *Pcgf3/5* were dispensable for embryonic stem cell self-renewal (Zhao et al, 2017; Yao et al, 2018). Here, we showed that *Pcgf3/5* regulate NSC proliferation.

In summary, a side-by-side comparison of double knockout of different *Pcgf* homologs revealed a stronger contribution of canonical PRC1.2/1.4 compared with non-canonical PRC1.3/1.5 to the regulation of NSC proliferation.

## Canonical, but not non-canonical, PRC1 regulates the differentiation potential of NSCs

Next, we aimed to dissect the contribution of canonical and non-canonical PRC1 to NSC differentiation. For this, we made use of the previously described potential of NSCs to maintain their developmental progression in vitro, initially resulting in the production of neurons (neurogenic phase), followed by the generation of astrocytes (gliogenic phase) (Hirabayashi et al, 2009). Deletion of *Pcgf2/4* in freshly isolated NSCs, which were induced to differentiate by the withdrawal of growth factors and the addition of serum to the medium, resulted in the generation of more neurons at the expense of oligodendrocytes, leaving the proportion of astrocytes unchanged (Fig 3A–C). In contrast, deletion of *Pcgf3/5* did not result in any significant changes in the proportions of differentiated cell types (Fig 3D and E). This suggests that canonical and non-canonical PRC1 complexes differentially contribute to the regulation of NSC fate during the neurogenic phase.

To compare the contribution of different PRC1 subcomplexes during the gliogenic phase, we kept the NSC lines in culture for 20 d and then repeated the differentiation experiment (Fig 3F). Although cKO of *Pcgf2/4* resulted in the generation of more

astrocytes compared with control, cKO of *Pcgf3/5* again did not affect the differentiation potential of NSCs (Fig 3G–J). This side-by-side comparison highlights the role of canonical PRC1.2/1.4, but not non-canonical PRC1.3/1.5, in determining the lineage potential of NSCs during differentiation.

Taken together, deletion of *Pcgf2/4* resulted in an increased proportion of neurons during the neurogenic phase and of astrocytes during the gliogenic phase. This is in contrast to the deletion of *Ring1b*, central to all PRC1 complexes, which was shown to not affect neuron numbers during the neurogenic phase, but resulted in more neurons during the gliogenic phase (Hirabayashi et al, 2009). These results underscore the differential contributions of PRC1 subcomplexes with different subunit compositions during neurogenic and gliogenic phases of neural differentiation.

## Canonical PRC1 regulates the expression of stem cell and neurogenic factors

Lastly, to explore the mechanism by which canonical PRC1 regulates NSC fate during differentiation, we performed gene expression analysis by RNA-seq, 4 d after induction of differentiation, followed by deletion of *Pcgf2/4* in neurogenic NSCs (Figs 4A and B and S2A–C). In line with the repressive function of PRC1, we identified 120 genes that were more than twofold up-regulated upon deletion of *Pcgf2/4* compared with control, whereas only 24 genes were down-regulated (Fig 4C). The differentially expressed genes (DEG) were characterized by gene ontology (GO) terms related to the molecular functions "DNA-binding" and "E-box-binding" (Fig 4D), which is in accordance with hallmarks of Polycomb-mediated regulation through binding of genes encoding key developmental transcription factors (Bernstein et al, 2006; Schuettengruber et al, 2017). Moreover, GO terms related to biological processes included "regionalization" and "pattern specification" (Fig 4E), extending what has been described for other Polycomb proteins during brain development (Hirabayashi et al, 2009; Albert et al, 2017; Eto et al, 2020). Of the 120 up-regulated genes, the majority (89 genes) were directly bound by Pcgf2 and/or Ring1b in NPCs derived from embryonic stem cells (Fig 4F) (Kloet et al, 2016). Among the direct PRC1 targets that were up-regulated, we found many transcription factors (Fig 4G), including *Hox* genes (Fig 4H), which represent known targets of PRC1 (Akasaka et al, 2001; Kloet et al, 2016).

Next, we aimed to specifically explore the expression of factors that may underlie the shifts in NSC lineage potential that we observed in *Pcgf2/4* cKO NSCs. We found that several stem cell factors, including *Id2*, *Pax6*, and *Hes5*, showed reduced expression after

**Figure 1. PRC1 components are differentially expressed in the human and mouse developing neocortex.**
**(A)** Schematic illustration of canonical and non-canonical PRC1 complexes containing the RING1A/B core and either PCGF2/4 or PCGF3/5 subunits, respectively.
**(B)** Schematic illustration of the human developing neocortex, divided into the ventricular zone (VZ), inner subventricular zone (ISVZ), outer subventricular zone (OSVZ), intermediate zone (IZ), and cortical plate (CP). **(C)** PRC1 mRNA expression levels in the human developing neocortex at gestation week (GW) 14 analysed by RNA-seq (data from Fietz et al [2012]), relative to the housekeeping gene *ACTB*. **(D)** Immunofluorescence for the radial glial marker SOX2, neuronal marker CTIP2, and PRC1-related RING1B, H2AK119ub1, PCGF2, and PCGF4 of the human foetal tissue (GW12/13). **(E)** Quantifications of mean intensity per cell in the indicated zones of the human foetal tissue, relative to the intensity in the VZ. **(F)** Schematic illustration of the mouse developing neocortex. **(G)** PRC1 mRNA expression levels in the mouse developing neocortex at E14.5 analysed by RNA-seq (data from Fietz et al [2012]), relative to the housekeeping gene *Actb*. **(H)** Immunofluorescence of the mouse embryonic tissue (E14.5). **(I)** Quantifications of mean intensity per cell in the indicated zones of the mouse embryonic tissue, relative to the intensity in the VZ. Data information: Scale bars, 100 μm. Bar graphs represent mean values. **(C, D, G, I)** Error bars represent the SD of (C, D) three tissue samples from different individuals and (G, I) 3–5 embryos from at least two different litters. ***P < 0.001, **P < 0.01, *P < 0.05, Tukey's multiple comparison test.

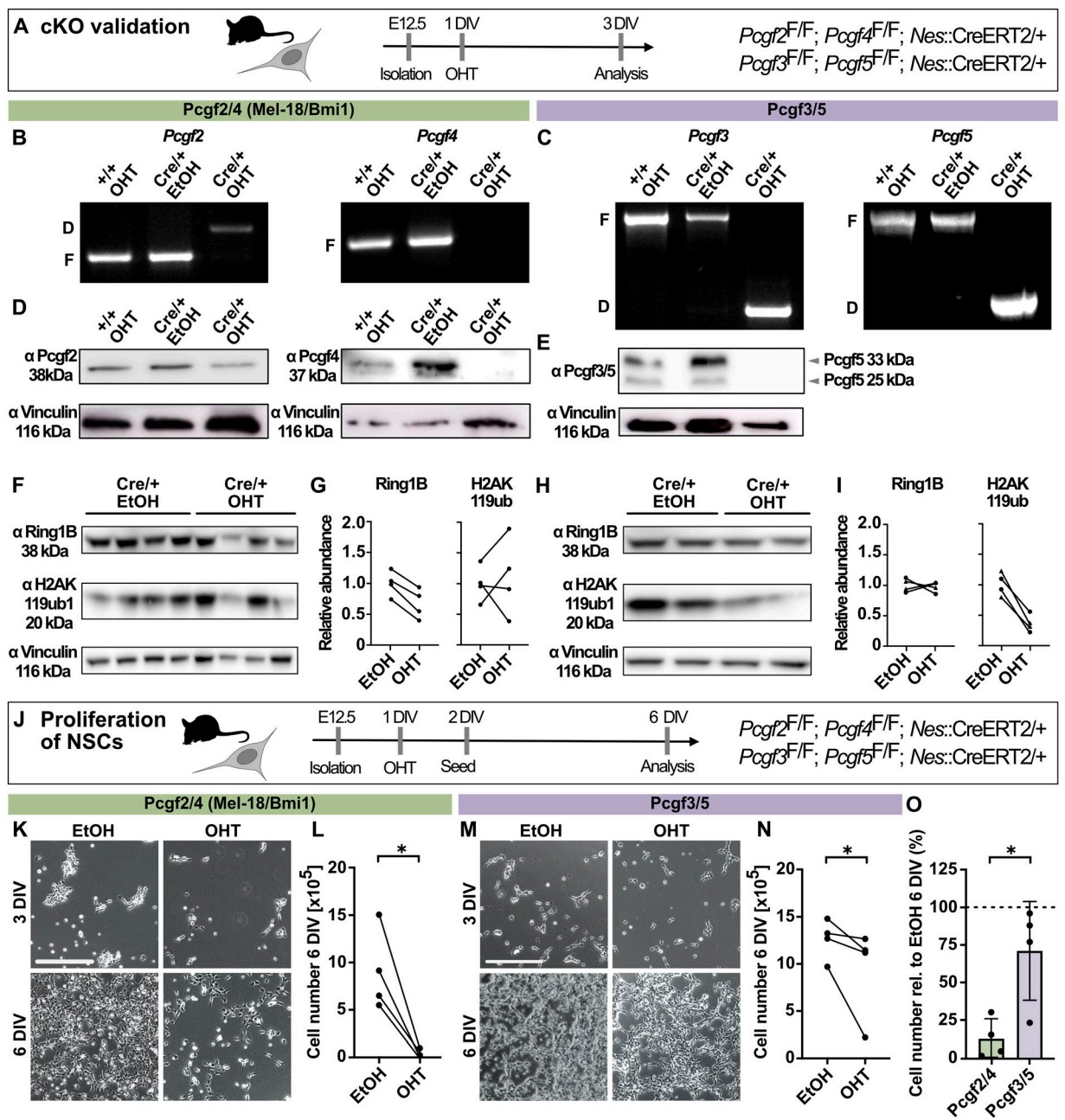

**Figure 2. Deletion of *Pcgf2/4* and *Pcgf3/5* reduces NSC proliferation to different degrees.**
**(A)** Schematic of experimental workflow. NSCs were isolated from E12.5 embryos from either *Pcgf2*F/F; *Pcgf4*F/F; *Nes*::CreERT2/+ or *Pcgf3*F/F; *Pcgf5*F/F; *Nes*::CreERT2/+ mouse lines. After 1 d in vitro (DIV), the deletion of Pcgf components was induced by the administration of 4-hydroxytamoxifen (OHT) and analysis was performed at 3 DIV. **(B, C)** Genotyping of *Pcgf2/4* (B) and *Pcgf3/5* (C) floxed ("F") and deletion ("D") alleles by PCR analysis after treatment of NSC cultures from control (+/+) or experimental (Cre/+) mice with ethanol ("EtOH"; control) or OHT. **(D, E)** Immunoblots of protein lysates from the same NSC cultures shown in (B), using anti-Pcgf2, anti-Pcgf4, anti-Pcgf3/5, and anti-vinculin antibodies. **(F)** Immunoblots of protein lysates from NSCs upon *Pcgf2/4* deletion, using anti-Ring1B, anti-H2AK119ub1, and anti-vinculin antibodies. **(F, G)** Quantitative analysis of protein abundance from blots in (F) for Ring1B and H2AK119ub1, relative to vinculin. **(H)** Immunoblots of protein lysates from NSCs upon *Pcgf3/5* deletion, using anti-Ring1B, anti-H2AK119ub1, and anti-vinculin antibodies. **(H, I)** Quantitative analysis of protein abundance from blots in (H) for Ring1B and H2AK119ub1, relative to vinculin. **(J)** Schematic of experimental workflow. Deletion of *Pcgf* genes was induced at 1 DIV, 50,000 cells were seeded at 2 DIV, and cells were counted at 6 DIV. **(K)** Brightfield images of *Pcgf2*F/F; *Pcgf4*F/F; *Nes*::CreERT2/+ NSC cultures treated with EtOH or OHT at 3 DIV and 6 DIV. **(L)** Quantification of cell numbers at 6 DIV. **(M)** Brightfield images of *Pcgf3*F/F; *Pcgf5*F/F; *Nes*::CreERT2/+ NSC cultures treated with EtOH or OHT at 3 DIV and 6 DIV. **(N)** Quantification of cell numbers at 6 DIV. **(H, J, O)** Data from (H, J) plotted for comparison of cell numbers after deletion of either *Pcgf2/4* or *Pcgf3/5*, shown as percentage relative to the EtOH control. Data information: Scale bars, 300 μm. Bar graphs represent mean values. Error bars represent the SD. **(G, I, L, N)** Dots connected by lines represent four embryos from at least two independent litters treated with either EtOH or OHT. **(G, I, L, N)** **P < 0.01, unpaired *t* test; (L, N) *P < 0.05, Tukey's multiple comparison test.
Source data are available for this figure.

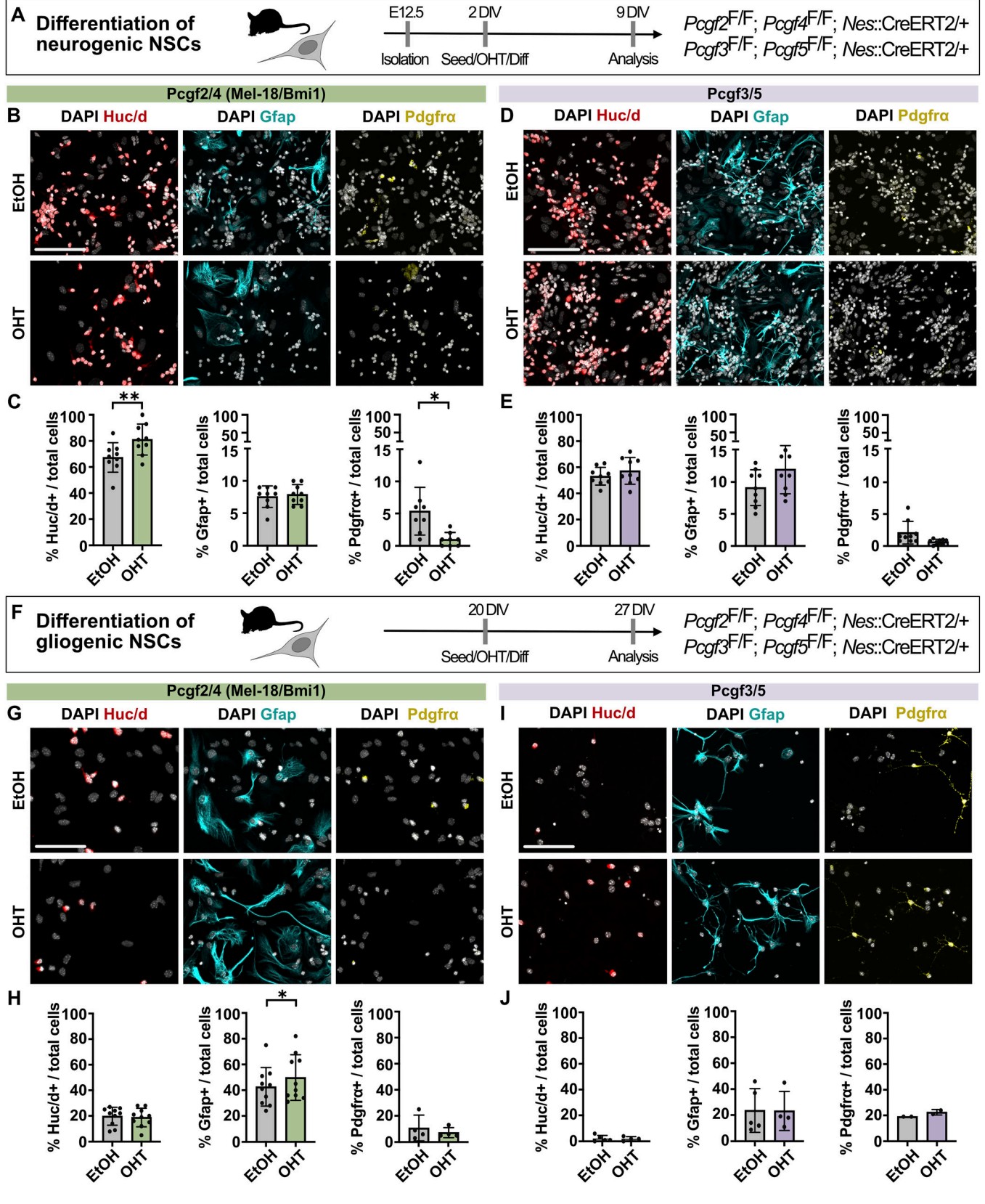

*Pcgf2/4* deletion, whereas neurogenic factors (*Lhx5*, *Lhx9*, *Nr4a2*) and neuronal maturation genes (*En1*, *En2*, *Pitx3*) were increased compared with control (Fig 5A–D). These genes were bound by Pcgf2 and/or Ring1b in NPCs (Fig 4F) (Kloet et al, 2016), suggesting that they may represent direct targets of canonical PRC1 and may contribute to the enhanced neuronal differentiation of neurogenic NSCs after deletion of *Pcgf2/4*. These conclusions were further supported by gene set enrichment analysis (Fig S2D), which revealed differences in the enrichment of gene sets associated with NPCs and neocortex development (Florio et al, 2015), in particular genes with high CpG promoters previously associated with PRC2-mediated H3K27me3 (Mikkelsen et al, 2007; Meissner et al, 2008).

In contrast, regulators of astrocyte fate, such as *Gfap*, *Hmgn1*, and *Hmgn2*, did not show altered expression (Fig 5E) and were not bound by Polycomb (Kloet et al, 2016; Albert et al, 2017), which is in agreement with previous reports suggesting that astrocyte-specific genes are regulated by DNA methylation in NPCs (Fan et al, 2005; Hatada et al, 2008). Several genes encoding oligogenic factors (*Olig2*, *Gli2*, *Sox9*) were directly bound by Pcgf2 and/or Ring1b in NPCs (Kloet et al, 2016) and showed a trend for reduced expression, even though not significantly (Fig 5F).

# Discussion

In summary, our side-by-side comparison of *Pcgf2/4* and *Pcgf3/5* deletion in NSCs revealed a differential contribution of canonical and non-canonical PRC1, respectively, to the regulation of NSC proliferation and lineage potential (Fig 5G). Despite the observation that Pcgf2/4 expression is highest in the neuronal population, we observed that canonical PRC1.2/1.4 contributes to the regulation of NSCs at proliferative, neurogenic, and gliogenic phases. Even though *Pcgf5* is preferentially expressed in the ventricular zone in vivo, deletion of *Pcgf3/5* only had a minor effect on NSC proliferation in vitro and did not impact NSC differentiation, during neither the neurogenic nor the gliogenic phase. It remains possible that additional non-canonical complexes (PRC1.1/1.6) may functionally contribute to the regulation of NSC fate, even though at least *Pcgf1* is expressed at low levels in the mouse developing neocortex (Fietz et al, 2012).

Mechanistically, PRC1 was reported to bind to several stem cell and neurogenic genes, which we found to be differentially expressed upon *Pcgf2/4* deletion, suggesting that these genes might be directly regulated by canonical PRC1. Our data suggest that rather than different PRC1 subcomplexes contributing to different phases of neural development, it is canonical PRC1.2/1.4 that regulates NSC proliferation and differentiation, whereas PRC1.3/1.5 plays a minor role in these processes. Overall, this suggests that switches in subunit composition may be more characteristic to the exit from pluripotency and differentiation towards different tissues and organs (Luis et al, 2011; Morey et al, 2012, 2015; O'Loghlen et al, 2012; Klauke et al, 2013; Santanach et al, 2017), but may be less relevant within a given lineage, such as neural development.

# Materials and Methods

**Reagents and tools table.**

| Reagent/resource | Reference or source | Identifier or catalogue number |
|---|---|---|
| Experimental models | | |
| GW12/13 human foetal brain tissue | This study | N/A |
| Mouse: NSC | This study | N/A |
| Mouse: *Pcgf2*$^{F/F}$; *Pcgf4*$^{F/F}$ | Fursova et al (2019) | N/A |
| Mouse: *Pcgf3*$^{F/F}$; *Pcgf5*$^{F/F}$ | Almeida et al (2017) | N/A |
| Mouse: *Nes*::CreERT2/+ | Imayoshi et al (2006) | N/A |
| Antibodies | | |
| Goat anti-Sox2 (IF 1:200) | AF2018; R&D Systems | RRID: AB_355110 |
| Rat anti-Ctip2 (IF 1:250) | ab18465; Abcam | RRID: AB_2064130 |
| Mouse anti-Ring1B (IF 1:1,000; WB 1:1,000) | 39663; Active Motif | RRID: AB_2716831 |
| Rabbit anti-Ubiquityl-Histone H2A (Lys119) (IF 1:1,500) | 8240; Cell Signaling | RRID: AB_10891618 |

**Figure 3. Deletion of *Pcgf2/4* but not *Pcgf3/5* results in altered lineage potential of neurogenic and gliogenic NSCs.**
**(A)** Schematic of experimental workflow. Deletion of *Pcgf* genes was induced at 2 DIV concomitant with seeding of NSCs with neurogenic potential and induction of differentiation. Differentiated cells were analysed at 9 DIV. **(B)** DAPI staining and immunofluorescence for the pan-neuronal marker HuC/D, the astrocyte marker Gfap, and the oligodendrocyte precursor marker PDGFRα after 7 d of differentiation of *Pcgf2*$^{F/F}$; *Pcgf4*$^{F/F}$; *Nes*::CreERT2/+ NSCs. **(C)** Quantification of the percentage of marker-positive cells of total DAPI-positive cells after deletion of *Pcgf2/4*. **(D)** DAPI staining and immunofluorescence after 7 d of differentiation of *Pcgf3*$^{F/F}$; *Pcgf5*$^{F/F}$; *Nes*::CreERT2/+ NSCs. **(E)** Quantification of the percentage of marker-positive cells of total DAPI-positive cells after deletion of *Pcgf3/5*. **(F)** Schematic of experimental workflow. Deletion of *Pcgf* genes was induced at 20 DIV in NSCs with gliogenic potential, concomitant with seeding and induction of differentiation. **(G)** DAPI staining and immunofluorescence for lineage markers. **(H)** Quantification of the percentage of marker-positive cells. **(I)** DAPI staining and immunofluorescence for lineage markers. **(J)** Quantification of the percentage of marker-positive cells. Data information: Scale bars, 100 μm. Bar graphs represent mean values. **(C, E, H, J)** Error bars represent the SD of (C, E, H, J) 4–9 embryos from at least two different litters. **P < 0.01, *P < 0.05, Tukey's multiple comparison test.

**Continued**

| Reagent/resource | Reference or source | Identifier or catalogue number |
|---|---|---|
| Mouse anti-Mel18 (IF 1:50, WB 1:200) | SC-515329; Santa Cruz | RRID: AB_2687587 |
| Mouse anti-Bmi-1 (IF 1:300; WB 1:1,000) | 05-637; EMD Millipore | RRID: AB_309865 |
| Chicken anti-Nestin (IF 1:500) | ab134017; Abcam | RRID: AB_2753197 |
| Mouse anti-HuC/HuD (IF 1:300) | A21271; Invitrogen | RRID: AB_221448 |
| Rat anti-GFAP (IF 1:500) | 13-0300; Invitrogen | RRID: AB_2532994 |
| Rabbit anti-PDGFR alpha (IF 1:500) | ab203491; Abcam | RRID: AB_2893014 |
| Mouse anti-APC (CC-1, IF 1:500) | OP80; Merck | RRID: AB_2057371 |
| Rabbit anti-PCGF3/5 (WB 1:1,000) | ab201510; Abcam | RRID: AB_2818981 |
| Mouse anti-vinculin (WB 1:1,000) | V9131; Sigma-Aldrich | RRID: AB_477629 |
| Donkey anti-goat IgG, Alexa Fluor 488 conjugated (1:1,000) | A-11055; Invitrogen | RRID: AB_2534102 |
| Goat anti-rat IgG, Alexa Fluor 633 conjugated (1:1,000) | A-21094; Invitrogen | RRID: AB_2535731 |
| Donkey anti-mouse IgG, Alexa Fluor 555 conjugated (1:1,000) | A-31570; Invitrogen | RRID: AB_2536180 |
| Donkey anti-rabbit IgG, Alexa Fluor 555 conjugated (1:1,000) | A-31572; Invitrogen | RRID: AB_2536182 |
| Goat anti-chicken IgG, Alexa Fluor 488 conjugated (1:1,000) | A-11039; Invitrogen | RRID: AB_2534096 |
| Donkey anti-rat IgG, Alexa Fluor 555 conjugated (1:1,000) | A-48270; Invitrogen | RRID: AB_2536114 |
| Donkey anti-mouse IgG, Alexa Fluor 647 conjugated (1:1,000) | A-31571; Invitrogen | RRID: AB_162542 |
| Donkey anti-rabbit IgG, Alexa Fluor 647 conjugated (1:1,000) | A-31573; Invitrogen | RRID: AB_2536183 |
| Goat anti-rabbit IgG, H&L HRP (1:3,000) | ab6721; Abcam | RRID: AB_955447 |
| Goat anti-mouse IgG, H&L HRP (1:10,000) | ab205719; Abcam | RRID: AB_2755049 |
| Oligonucleotides | | |
| Primers | This study | Table S1 |
| Chemicals, enzymes, and other reagents | | |
| DMEM/F-12, Hepes | Gibco | Cat. # 31330095 |
| Poly-D-lysine | Invitrogen | Cat. # A3890401 |
| Laminin from the Engelbreth–Holm–Swarm murine sarcoma basement membrane | Sigma-Aldrich | Cat. # L2020 |
| BSA | Sigma-Aldrich | Cat. # A2153 |
| FBS | Sigma-Aldrich | Cat. # F7524 |
| Heparin sodium salt from porcine intestinal mucosa | Sigma-Aldrich | Cat. # H4784 |
| MEM Non-essential Amino Acid Solution (100X) | Gibco | Cat. # 11140050 |
| GlutaMAX supplement | Gibco | Cat. # 35050038 |
| Human EGF Recombinant Protein | Gibco | Cat. # PHG0311 |
| Human FGF-basic (FGF-2/bFGF) (aa 10-155) Recombinant Protein | Gibco | Cat. # PHG0021 |
| B-27 supplement (50x), serum-free | Gibco | Cat. # 17504044 |
| N-2 supplement (100X) | Gibco | Cat. # 17502048 |
| OH-tamoxifen | Sigma-Aldrich | Cat. # H7904 |
| Thermo Fisher Scientific Halt Protease Inhibitor Cocktail (100x) | Thermo Fisher Scientific | Cat. # 10516495 |

**Continued**

| Reagent/resource | Reference or source | Identifier or catalogue number |
|---|---|---|
| DAPI | Roche | Cat. # 10236276001 |
| Pierce Detergent Compatible Bradford Assay Kit | Thermo Fisher Scientific | Cat. # 23246 |
| PVDF transfer membranes, 0.45 µm | Thermo Fisher Scientific | Cat. # 88518 |
| SuperSignal West Pico PLUS Chemiluminescent Substrate | Thermo Fisher Scientific | Cat. # 34579 |
| Software and algorithms | | |
| Fiji/ImageJ | Fiji/ImageJ | https://imagej.nih.gov/ij/ |
| Prism (8.4.3) | GraphPad Software | N/A |
| Geneious Prime (2019.2.1) | Biomatters Ltd. | N/A |
| Affinity Photo + Designer (1.10.5.1342) | Serif Ltd. | N/A |
| FACSDiva (8.0.2) | BD Biosciences | N/A |
| FastQC (v0.11.6) | Babraham Bioinformatics | https://www.bioinformatics.babraham.ac.uk/projects/fastqc/ |
| Other | | |
| RNeasy Mini Kit | QIAGEN | Cat. # 74104 |
| QIAquick Gel Extraction Kit | QIAGEN | Cat. # 28106 |
| Quick-DNA/RNA Microprep Plus Kit | Zymo Research | Cat. # D7005 |
| Raw and analysed RNA-seq data | This study | NCBI GEO: GSE274747 |
| RNA-seq | Fietz et al (2012) | NCBI GEO: GSE38805 |
| ChIP-seq Ring1b | Kloet et al (2016) | NCBI GEO: GSM1917303 |
| ChIP-seq Pcgf2 | Kloet et al (2016) | NCBI GEO: GSM1917304 |
| Phylogenetic pictures | N/A | https://www.phylopic.org/ |
| Phylogenetic pictures | Shutterstock | N/A |

**Mice**

All experimental procedures were conducted in agreement with the German Animal Welfare Legislation after approval by the Landesdirektion Sachsen (licences DD24.1-5131/476/8; 25-5131/521/16). Animals were kept on a 12-h/12-h light/dark cycle with food and water ad libitum. Mice used for PRC1 expression analysis were wild-type mice from the inbred C57BL/6J strain. Embryonic day (E) 0.5 was set as noon on the day on which the vaginal plug was observed. All experiments were performed in the dorsolateral telencephalon of mouse embryos, at a medial position along the rostro-caudal axis. The developmental time point E14.5 of experimental procedures corresponds to a mid-neurogenic stage, when the production of upper-layer neurons has started. The sex of embryos was not determined, as it is not likely to be of relevance to the results obtained in the present study.

For the inducible deletion of *Pcgf* genes in NSCs, *Pcgf2*$^{F/F}$; *Pcgf4*$^{F/F}$ (Fursova et al, 2019) and *Pcgf3*$^{F/F}$; *Pcgf5*$^{F/F}$ (Almeida et al, 2017) mice were crossed with *Nes*::CreERT2/+ mice (Imayoshi et al, 2006). NSCs were isolated from either *Pcgf2*$^{F/F}$; *Pcgf4*$^{F/F}$; *Nes*::CreERT2/+ or *Pcgf3*$^{F/F}$; *Pcgf5*$^{F/F}$; *Nes*::CreERT2/+ strains. The genotyping primers are listed in Table S1. Only male embryos were included for *Pcgf3*$^{F/F}$;

*Pcgf5*$^{F/F}$; *Nes*::CreERT2/+ to exclude effects attributed to changes in X chromosome inactivation (Almeida et al, 2017).

**Human foetal brain tissue**

The human foetal brain tissue was obtained from the Department of Gynaecology and Obstetrics, University Clinic Carl Gustav Carus of the Technische Universität Dresden, after elective pregnancy termination and informed written maternal consent, and with approval of the local University Hospital Ethical Review Committee (IRB00001473; IORG0001076; ethical approval number EK 355092018), in accordance with the Declaration of Helsinki. The age of foetuses ranged from gestation weeks (GW) 12 to 13 as assessed by ultrasound measurements of crown–rump length and other standard criteria of developmental stage determination. The developmental time point corresponds to an early/mid-neurogenic stage, when the OSVZ expands and the production of upper-layer neurons starts. Because of protection of data privacy, the sex of the human foetuses, from which the neocortex tissue was obtained, cannot be reported. The sex of the human foetuses is not likely to be of relevance to the results obtained in the present study. The foetal neocortex tissue samples used in this study reported no health disorders. The foetal human brain tissue was dissected in Tyrode's solution and fixed immediately (within 1 h).

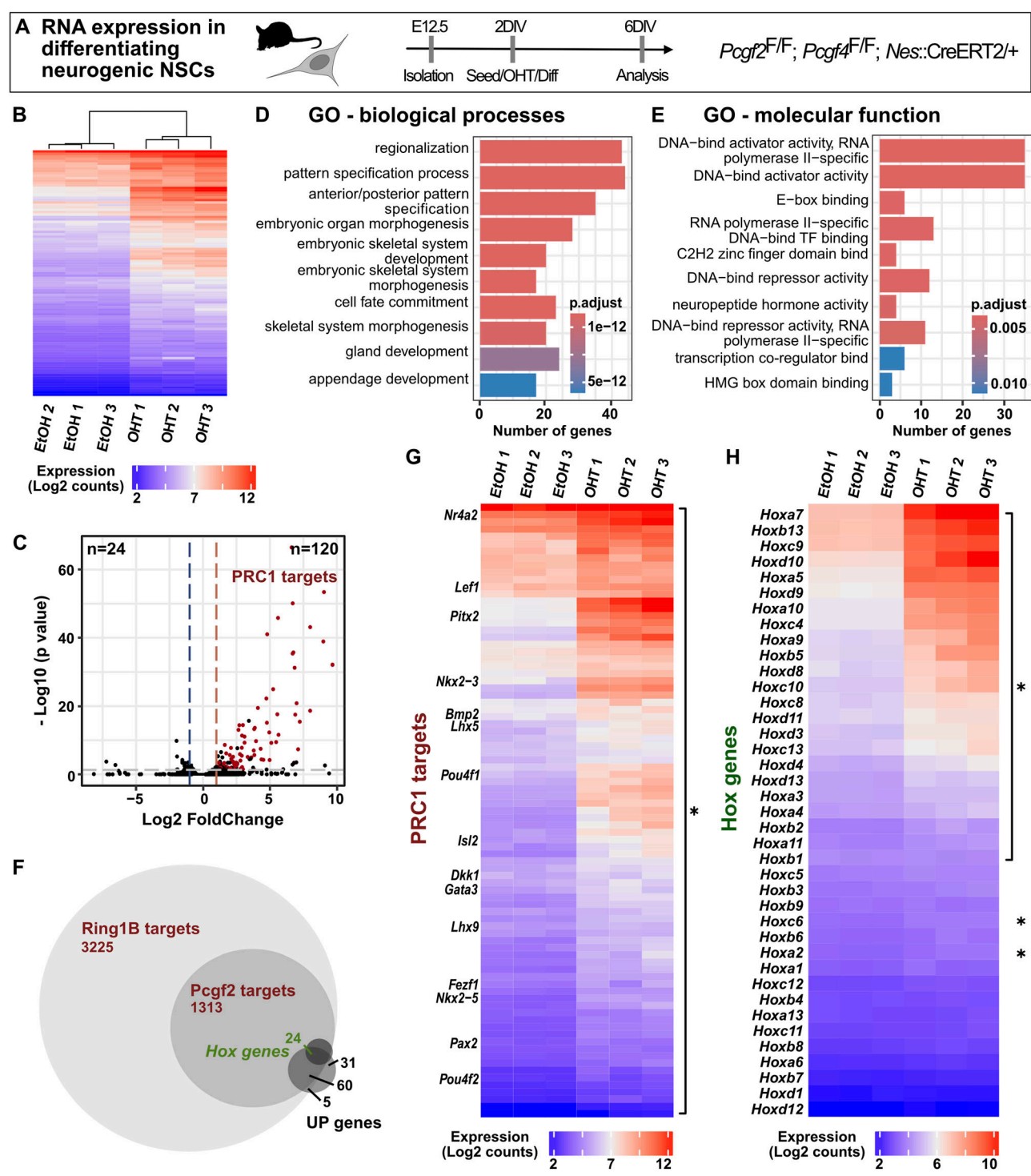

**Figure 4. Deletion of *Pcgf2/4* results in derepression of PRC1 target genes linked to fate specification.**
**(A)** Schematic of experimental workflow. Deletion of *Pcgf2/4* genes was induced at 2 DIV concomitant with seeding of NSCs with neurogenic potential and induction of differentiation. Gene expression was analysed by RNA-seq after 4 d of differentiation at 6 DIV. **(B)** Hierarchical clustering analysis with the heatmap of the 100 most differentially expressed genes between control (EtOH) and *Pcgf2/4* deletion (OHT) samples, showing the clustering of replicates. **(C)** Volcano plot of log10 (*P*-value) against log$_2$ fold change representing the differences in gene expression between control samples (EtOH) and *Pcgf2/4* cKO (OHT). The grey line represents the cut-off of *P* < 0.05; the blue line, the cut-off of log$_2$ fold change < −1; and the red line, the cut-off of log$_2$ fold change > 1. Numbers in the upper corners indicate the number of significantly up- and down-regulated genes, respectively. Genes bound by Ring1b and/or Pcgf2 in NPCs (Kloet et al, 2016) are highlighted ("PRC1 targets"). **(D, E)** Gene ontology (GO)–term enrichment analysis of up-regulated genes (*P* < 0.05) was performed for biological processes **(D)** and molecular function **(E)**. **(F)** Venn diagram representing the overlap of genes bound by Ring1b and Pcgf2 in NPCs (Kloet et al, 2016) with up-regulated genes, which include several *Hox* genes. **(G)** Heatmap of the genes up-regulated after *Pcgf2/4* deletion (OHT) and bound by Ring1b and Pcgf2 in NPCs (Kloet et al, 2016), with a log$_2$ fold change >1. **(H)** Heatmap of up-regulated *Hox* genes, with a log$_2$ fold change >1. Data information: Replicates represent NSCs from three embryos from two different litters.

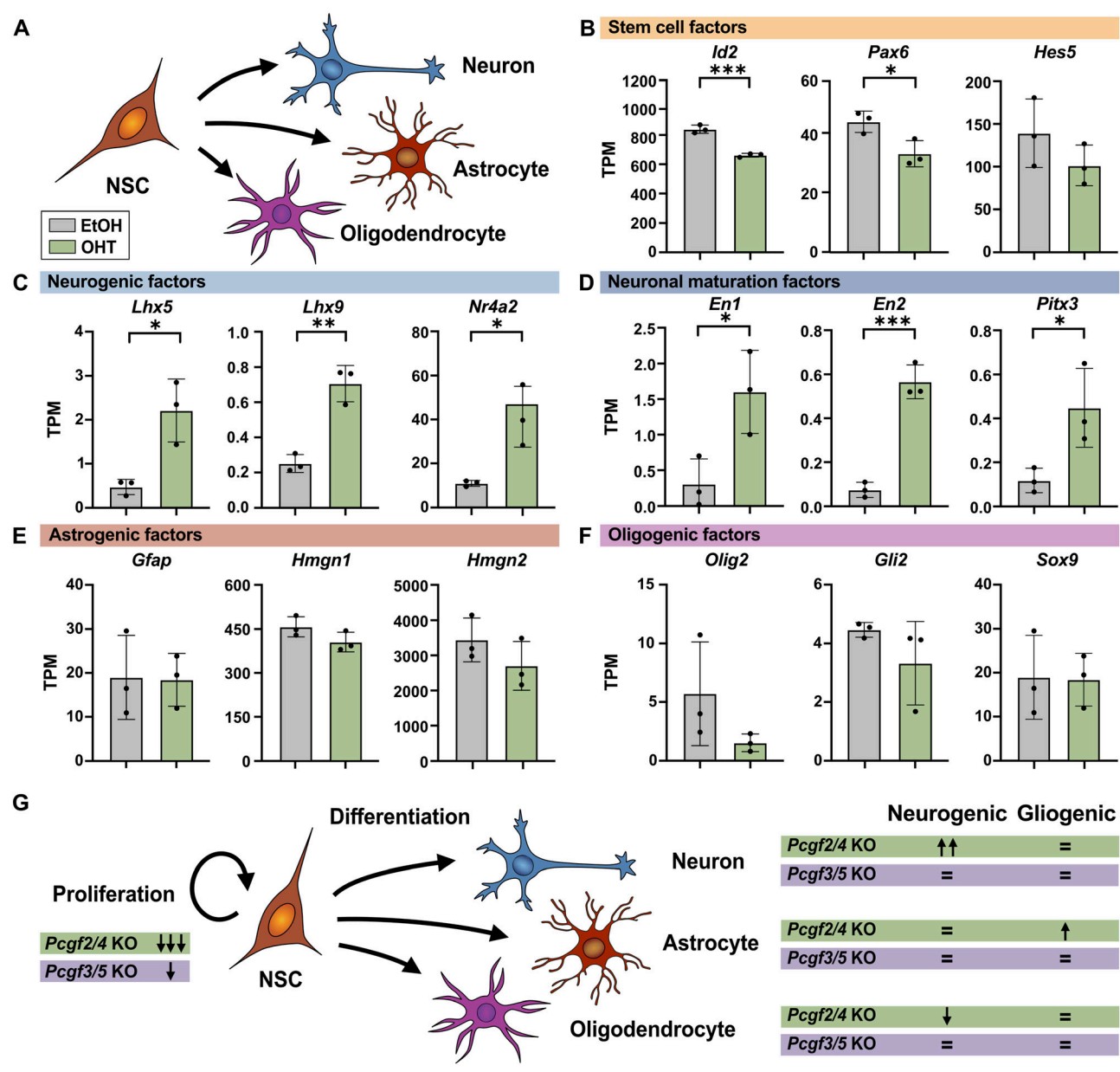

**Figure 5. Deletion of *Pcgf2/4* affects the expression of stem cell and neurogenic genes.**
**(A)** Schematic of the NSC differentiation paradigm. **(B, C, D, E, F)** mRNA expression in differentiating neurogenic NSCs (as described in Fig 4) in transcripts per million (TPM) analysed by RNA-seq for genes encoding (B) stem cell, (C) neurogenic, (D) neuronal maturation, (E) astrogenic, and (F) oligogenic factors. Deletion of *Pcgf2/4* genes was induced with OHT (green), whereas control cells were treated with EtOH (grey). **(G)** Summary of the functional role of *Pcgf2/4* and *Pcgf3/5* in the regulation of NSC proliferation and differentiation during the neurogenic and gliogenic phase. Data information: Bar graphs represent mean values. Error bars represent the SD of NSCs from three embryos from two independent litters. ***$P < 0.001$, **$P < 0.01$, *$P < 0.05$, unpaired $t$ test.

## Mouse NSC culture and differentiation

Mouse NSCs were isolated from E12.5 embryos as previously described (Schmitz et al, 2011; Schütze et al, 2022; Cubillos et al, 2024). Briefly, the dorsolateral cortex was isolated from E12.5 mice and treated with trypsin–EDTA (#25300054; Gibco) at 37°C for 20 min. Subsequently, soybean trypsin inhibitor (0.25 mg/ml in PBS, #17075-029; Invitrogen) was added, and cells were mechanically dissociated by pipetting and pelleted at 300$g$ for 5 min. NSCs were then plated on poly-D-lysine (at least 2 h at 37°C; #A3890401; Gibco)– and laminin (at least 4 h at 37°C; #L2020; Sigma-Aldrich)-coated plates at a density of 40,000 cells/mm$^2$ and cultured under standard conditions (37°C, 5% $CO_2$). Culture medium was prepared as a 1:1 mixture of DMEM/F-12 and Neurobasal medium (#12348-017; Gibco) supplemented with 10 ng/ml EGF, 20 ng/ml FGF, 1X N-2 and 1X B-27 supplements, 1X penicillin–streptomycin, 1X sodium pyruvate, 1X GlutaMAX, 1X MEM-NEAA, 4 mg/ml heparin, 5 mM Hepes, 0.01 mM 2-mercaptoethanol, and 100 mg/litre BSA. Differentiation of NSCs was

induced with medium deficient of EGF and FGF, supplemented with 2% FBS (#F7524; Sigma-Aldrich). NSCs were considered neurogenic in the first 5 DIV and gliogenic for the following passages as described before (Hirabayashi et al, 2009). Differentiation assays of neurogenic NSCs were performed at 2 DIV and of gliogenic NSCs after passaging every 3–4 d at 20 DIV. For immunohistochemistry analysis, NSCs were plated on poly-D-lysine– and laminin-coated coverslips (#0111520; Marienfeld-Superior) in a 24-well plate, with either 10,000 cells per well for the NSC proliferation assay or 50,000 cells per well for the differentiation assay. The medium was changed every second day. Deletion of *Pcgf* genes was induced by the addition of 1 $\mu$M 4-hydroxytamoxifen (OHT) to culture medium for 24–48 h, as previously described (Fursova et al, 2019). 4-Hydroxytamoxifen was dissolved in EtOH at a concentration of 1 mM and stored for a maximum of 4 mo at –20°C.

## Immunohistochemistry analysis of tissue sections

The tissue was fixed in 4% PFA in 120 mM phosphate buffer, pH 7.4, for 24 h at 4°C, washed twice in PBS, transferred to 30% sucrose for 24 h, embedded in O.C.T. compound (#4583; Sakura Finetek) with 15% sucrose, and frozen on dry ice. The tissue was cut into 12-$\mu$m cryosections on a Thermo Fisher Scientific NX70 cryostat. Immunofluorescence was performed as previously described (Florio et al, 2015; Cubillos et al, 2024). Antigen retrieval for 1 h with 10 mM citrate buffer, pH 6.0, at 70°C in a water bath was followed by three washes with PBS, quenching for 30 min in 0.1 M glycine in PBS, and blocking for 30 min in blocking solution (10% horse serum and 0.1% Triton in PBS) at RT. Primary antibodies were incubated in blocking solution overnight at 4°C. Subsequently, sections were washed three times in PBS, incubated with secondary antibodies (1:1,000) and DAPI (1:1,000) in blocking solution for 1 h at RT, and washed again three times in PBS before mounting on microscopy slides with Mowiol.

Images were acquired with a Zeiss ApoTome.2 fluorescence microscope with a 20x objective using 1.5-$\mu$m-thick optical sections. ZEN software was used to generate maximum intensity projections. To quantify the intensity of PRC1 markers, the human foetal and mouse embryonic tissue was divided into the different germinal zones and cortical plate based on SOX2 and CTIP2 staining and alignment of cells in DAPI within a 100-$\mu$m-wide image. The Fiji plug-in StarDist 2D (Schmidt et al, 2018) was applied using the versatile model to segment nuclei in DAPI, and if required, segmentation was corrected manually. Mean grey values of segmented nuclei were measured in Fiji and the resulting data processed using Excel and Prism software. The normal distribution of data was tested by Kolmogorov–Smirnov and Shapiro–Wilk tests, followed by Tukey's multiple comparison test.

## Immunohistochemistry analysis of NSC differentiation

Cells were fixed in 2% PFA in 120 mM phosphate buffer, pH 7.4, for 10 min at RT, before PFA was washed away twice with PBS. Cells were permeabilized with 0.1% Triton in PBS for 5 min, followed by washing twice with PBS for 5 min and twice with washing solution containing 0.1% Tween in PBS for 5 min. After this, cells were blocked for 30 min in blocking buffer containing 2.5% BSA (#A2153; Sigma-Aldrich), 0.1% Tween, and 10% horse serum in PBS, before

primary antibodies diluted in blocking buffer were added and incubated overnight at 4°C. This was followed by three washes with washing solution for 10 min, incubation with secondary antibodies (1:1,000) and DAPI (1:1,000) in blocking buffer for 1 h at RT, three additional washes, and embedding in a drop of Mowiol on microscopy slides.

Images were acquired with a Zeiss ApoTome.2 fluorescence microscope with a 20x objective using 1.5-$\mu$m-thick optical sections. ZEN software was used to generate maximum intensity projections. Samples were blinded after staining, before acquisition of images. For quantification of marker-positive cells, nuclei were segmented in DAPI using Fiji plug-in StarDist 2D (Schmidt et al, 2018) and counted with the cell counter tool in Fiji. The resulting data were processed using Excel and Prism software. Data were analysed for outliers using Grubb's test, followed by Tukey's multiple comparison test.

## Protein expression analysis by Western blotting

Proteins were isolated from cells in culture using TOPEX Plus buffer containing 300 mM NaCl, 50 mM Tris–HCl, pH 7.5, 0.5% Triton, 1% SDS, 1 mM DTT (#10708984001; Roche), 1X protease inhibitor (#4693116001; Roche), and 333.33 U/ml Benzonase (#E1014-25KU; Sigma-Aldrich) in water, by incubation at RT until viscosity disappeared (5–15 min). The protein concentration was measured using Pierce Detergent Compatible Bradford Assay Kit (#23246; Thermo Fisher Scientific). Subsequently, 20–40 $\mu$g of protein was resolved on a 10% SDS–PAGE gel and transferred to a PVDF transfer membrane (#88518; Thermo Fisher Scientific). Membranes were blocked for 1 h at RT with 5% skim milk in PBS with 0.05% Tween, and then incubated with primary antibodies overnight at 4°C. Secondary antibodies were incubated for 1 h at RT. Antibody signal was detected using the SuperSignal West Pico PLUS kit (#34579; Thermo Fisher Scientific).

## RNA-seq library preparation

RNA-seq was performed as previously described (Cubillos et al, 2024). RNA of differentiating NSCs was isolated using the Quick-RNA MiniPrep kit (#R1008; Zymo Research). Transcriptome libraries were prepared using an adapted version of the Smart-seq2 protocol (Picelli et al, 2013). Isolated total RNA from an equivalent of one 24-well plate was denatured for 3 min at 72°C in 4 $\mu$l hypotonic buffer (0.2% Triton X-100) in the presence of 2.4 mM dNTP, 240 nM dT primer (Table S1), and 4 U RNase inhibitor (M0314L; NEB). Reverse transcription was performed at 42°C for 90 min after filling up to 10 $\mu$l with RT buffer mix for a final concentration of 1X SuperScript II buffer (Invitrogen), 1 M betaine, 5 mM DTT, 6 mM MgCl$_2$, 1 $\mu$M TSO primer (Table S1), 9 U RNase inhibitor, and 90 U SuperScript II. The reverse transcriptase was inactivated at 70°C for 15 min. For subsequent PCR amplification of the cDNA, the optimal PCR cycle number was determined with an aliquot of 1 $\mu$l unpurified cDNA in a 10 $\mu$l qPCR containing 1X KAPA HiFi HotStart ReadyMix (Roche), 1X SYBR Green, and 0.2 $\mu$M UP primer (Table S1). The residual 9 $\mu$l cDNA was subsequently amplified using KAPA HiFi HotStart ReadyMix (Roche) at a 1X concentration together with 250 nM UP primer (Table S1) under the following cycling conditions: initial denaturation at 98°C for 3 min, 22 cycles (98°C for 20 s, 67°C for 15 s, and 72°C for

6 min), and final elongation at 72°C for 5 min. Amplified cDNA was purified using 1X volume of Sera-Mag SpeedBeads (GE Healthcare) resuspended in a buffer consisting of 10 mM Tris, 20 mM EDTA, 18.5% (wt/vol) PEG 8000, and 2 M sodium chloride solution. The cDNA quality and concentration were determined using Fragment Analyzer (Agilent).

For library preparation, 2 $\mu$l amplified cDNA was tagmented in 1X tagmentation buffer using 0.8 $\mu$l bead-linked transposome (Illumina DNA Prep, (M) Tagmentation, Illumina) at 55°C for 15 min in a total volume of 4 $\mu$l. The reaction was stopped by adding 1 $\mu$l of 0.1% SDS (37°C, 15 min). Magnetic beads were bound to a magnet, the supernatant was removed, and beads were resuspended in 14 $\mu$l indexing PCR Mix containing 1X KAPA HiFi HotStart ReadyMix (Roche) and 700 nM unique dual indexing primers (i5 and i7), and subjected to a PCR (72°C for 3 min, 98°C for 30 s, 12 cycles [98°C for 10 s, 63°C for 20 s, and 72°C for 1 min], and 72°C 5 min). Libraries were purified with 0.9X volume Sera-Mag SpeedBeads, followed by a double-size selection with 0.6X and 0.9X volume of beads, and a final 0.9X purification to obtain a fragment size distribution of 200–700 bp. Sequencing was performed after quantification using Fragment Analyzer on Illumina NovaSeq 6000 in 100-bp paired-end XP mode with an average sequencing depth of 40 million fragments per library.

### RNA-seq data analysis

Quality control of the sequencing data was performed with FastQC (version 0.11.9). Kallisto (version 0.64.1) (Bray et al, 2016) was used to align trimmed reads to mouse GRCm39. For further processing, data were imported into R using Tximport (Soneson et al, 2015) and EnsDb.Mmusculus.v79 packages. Raw fragment normalization based on library size and testing for differential expression between genotypes was performed with DESeq2 (version 1.30.1; Wald's test) (Love et al, 2014) with a false discovery rate (FDR) of 5% and a $\log_2$ fold change threshold of 1. To quantify gene expression levels within samples, transcripts per million (TPM) values were calculated with Tximport (Soneson et al, 2015). Volcano plots and heatmaps were generated with the R packages ggplot2 (Wickham et al, 2016) and Complexheatmap (Gu et al, 2016). Venn diagrams were generated using the online platform DeepVenn.com (Hulsen, 2022 *Preprint*). Differentially expressed genes were analysed by the KEGG pathway (https://www.genome.jp/), gene ontology (https://geneontology.org/) (Ashburner et al, 2000; Thomas et al, 2022), and gene set enrichment analysis (GSEA) (Subramanian et al, 2005). GSEA was performed using the Molecular Signature Database (Liberzon et al, 2015) C2 curated gene set.

### Statistical analysis

Sample sizes are reported in each figure legend. Sample sizes were estimated based on previous literature (Schmitz et al, 2011; Albert et al, 2017; Cubillos et al, 2024). All statistical analysis was performed using Prism (GraphPad Software). The normal distribution of datasets was tested by the Shapiro–Wilk or Kolmogorov–Smirnov test. Data were analysed for outliers using Grubb's test. The tests used included a $t$ test and Tukey's multiple comparison test, as indicated in the figure legend for each quantification. Significant changes are indicated by stars for each graph and described in the figure legends.

## Data Availability

RNA-seq data have been deposited with the Gene Expression Omnibus under the accession code GSE274747.

## Supplementary Information

## Acknowledgements

We are grateful to the facilities of the CRTD and DRESDEN-concept partner institutions for the outstanding support provided, notably R Hans from the Light Microscopy Facility, the teams for animal husbandry and histology, and the DRESDEN-concept Genome Center laboratory team for technical support. We thank all members of the Albert laboratory for help and discussions. We acknowledge R Kageyama from Kyoto University for providing the *Nes*::CreERT2 mouse line. M Albert acknowledges funding from the Center for Regenerative Therapies TU Dresden, the DFG (Emmy Noether, AL 2231/1-1), and the Schram Foundation.

### Author Contributions

J Hoffmann: conceptualization, formal analysis, investigation, visualization, and writing—review and editing.
TM Schütze: formal analysis and writing—review and editing.
A Kolodziejczyk: resources, investigation, and writing—review and editing.
K Küster: investigation.
A Kränkel: investigation.
S Reinhardt: resources and writing—review and editing.
RP Derihaci: resources.
C Birdir: resources.
P Wimberger: resources.
H Koseki: resources.
M Albert: conceptualization, supervision, funding acquisition, and writing—original draft, review, and editing.

### Conflict of Interest Statement

The authors declare that they have no conflict of interest.

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
