## [Reviewer comments · Life Science Alliance]

Life Science Alliance

Canonical and non-canonical PRC1 differentially contribute to regulation of neural stem cell fate

Janine Hoffmann, Theresa Schütze, Annika Kolodziejczyk, Karolin Küster, Annekathrin Kraenkel, Susanne Reinhardt, Razvan Derihaci, Cahit Birdir, Pauline Wimberger, Haruhiko Koseki, and Mareike Albert

DOI: <https://doi.org/10.26508/lsa.202403006>

Corresponding author(s): Mareike Albert, TU Dresden

Review Timeline:

Submission Date:	2024-08-21
Editorial Decision:	2024-09-27
Revision Received:	2025-01-27
Editorial Decision:	2025-01-28
Revision Received:	2025-01-30
Accepted:	2025-01-31

Transaction Report:

September 27, 2024

Re: Life Science Alliance manuscript #LSA-2024-03006-T

Dr. Mareike Albert
TUD Dresden University of Technology
Center for Regenerative Therapies Dresden (CRTD)
Fetscherstr. 105
Dresden 01307
Germany

Dear Dr. Albert,

Thank you for submitting your manuscript entitled "Canonical and non-canonical PRC1 differentially contribute to the regulation of neural stem cell fate" to Life Science Alliance. The manuscript was assessed by expert reviewers, whose comments are appended to this letter. We invite you to submit a revised manuscript addressing the Reviewer comments.

Thank you for this interesting contribution to Life Science Alliance. We are looking forward to receiving your revised manuscript.

Sincerely,

B. MANUSCRIPT ORGANIZATION AND FORMATTING:

Reviewer #1 (Comments to the Authors (Required)):

The manuscript by Matsuda and colleagues compares the role of canonical and non-canonical PRC1 complexes in NSC maintenance and differentiation. The topic is interesting and while the literature is rich about the roles of Polycomb in several system, this is an open question in the field.

The experimental approaches are elegant even though sometimes, in my vision not solid enough to draw conclusions (see below).

Thus, while the story is interesting and of worth there are missing points and limitations that at this stage dampen the enthusiasm for the manuscript in this form.

Major points

- The experiments regarding NSC differentiation in my vision are not solid as required to be one the key data of the work. Figure 3F-J: Differentiation in the control case (EtOH) should be equal in both lines (gray bars in H and J) but here it does not seem the case. In particular the Pcgf3/5 line is less efficient. Considering the small phenotype that is present in Pcgf2/4 line, is the poor differentiation of Pcgf3/5 line maybe masking the phenotype?
- I would suggest to perform additional experimental approaches to verify the findings. In utero electroporation of the Cre in different time windows to evaluate neuro- and gliogenesis? Differentiation in human setting?
- The additional setting will be also useful to validate the findings of the genomic analysis in a more compelling situation.
- Regarding the RNA-seq experiment: I would suggest to compare gene expression at the day 4 of differentiation with the starting point. In this way one can evaluate at the molecular level the neurogenic induction. Moreover, the temporal profile of expression will be evaluated on these important genes (either that should be upregulated or kept low during differentiation). Considering the dynamic role and activity of Polycomb complexes this may underline specific and direct PRC1 role during differentiation.
- GSEA analysis may return more complete picture than GO in this context. GSEA regarding "neuron differentiation" or "cortical development" should be presented.

Minor

- Please use the correct international nomenclature for genes and proteins.

Reviewer #2 (Comments to the Authors (Required)):

The manuscript by Hoffman et al. reports in vitro observations about proliferation and differentiation abilities of primary neural progenitors (NPCs) from wild type and mutant mouse lines that can be made double deficient in Polycomb PCGF subunits, either Pcgf2 and Pcgf4 or Pcgf3 and Pcgf5. The work tries to determine whether canonical and variant (non-canonical) Polycomb Repressive Complexes of class I (PRC1) regulate distinctly the developmental potential of NPCs. PCGF2 and PCGF4 form heterodimers with RING1A or RING1B in canonical PRC1 complexes, whereas PCGF3 and PCGF5 are present, together with the RING1 paralogs, in some of the variant PRC1 complexes. Mutant NPCs proliferative activity was reduced, compared to wild type cells, in both situations although to a large extent in double Pcgf2/Pcgf4 mutants. Likewise, larger differentiation alterations were observed for Pcgf2/Pcgf4 KO NPCs than for Pcgf3/Pcgf5 KO cells. Finally, transcriptome analysis of Pcgf2/Pcgf2 KO cells showed gene expression changes enriched within GO terms expected when dealing with Polycomb targets.

Although the nature of Polycomb complexes displayed by distinct cell states within given cell lineages is far from being well characterised, the fragmented evidence available support the hypothesis of the authors. I only have a reservation, also mentioned by the authors, that by not considering variant PRC1 complexes containing PCGF paralogs PCGF1 and PCGF6 is hard to extend the conclusions that suggest that canonical PRC1s are functionally more relevant to the developmental potential of NPCs than variant PRC1 complexes. In this regard, it would be beneficial to determine PCGF1 and PCGF6 protein expression in fetal brain sections so that indeed could be stated that PCGF3 and PCGF5-containing PRC1 complexes made most of the activity of variant PRC1s in developing brain.

Another concern is about the in vivo relevance of the observations in tissue culture experiments. This is important because of

the seemingly weak differences. I would imagine that this should be feasible given the availability of mouse lines combining the inducible alleles and a tissue-specific recombinase.

Regarding transcriptional alterations in mutant cells I don't see why Pcgf3/Pcgf5-mutant NPCs have been excluded from the analysis. Even if little changes could be anticipated, based on available evidence, which nevertheless points at observable phenotypes.

It would also be interesting to determine global RING1A and RING1B protein levels in Pcgf2/Pcgf4 KO NPCs. On the other hand, given the differential contribution of H2AK119Ub during neurogenesis (important at early stages, not so important at late stages) it would also be convenient to document it in the present set of experiments.

Other points

- Figure 2I-K: there is a lack of correspondence between tissue culture images (I) and plots (J,K). Three of the experiments indicate little differences between OHT- and EtOH-treated cultures but the images seem to show otherwise.
- Figure 3: Why the proliferative defects of Pcgf2/Pcgf4 KO cells are much milder than what one could expect from those suggested by Figure 2? Also, why EtOH-treated Pcgf2/Pcgf4 and EtOH-treated Pcgf3/Pcgf5 cultures (i.e. controls close to wild type) differ in percentages of Huc/d+ cells?. Differences are within the range seen between EtOH- and OHT-treated Pcgf2/Pcgf4 cells which is interpreted as functionally relevant.
- Figure 4F: I am sure that NPCs derived from embryonic stem cells and primary NPCs derived from fetal brain are probably similar. However, it might have been preferable to look for chromatin occupancy data from primary NPCs so that panel F can be said truly relevant.
- Figure 5: a more generous legend is necessary, in particular to the origin of data plotted in panels B-F.

Minor

Figure 1H lacks magnification bar.

The paper by Fasano et al. 2007 has been retracted by the authors.

Response to Reviewers

Reviewer #1

Reviewer's comment:

The manuscript by Matsuda and colleagues compares the role of canonical and non-canonical PRC1 complexes in NSC maintenance and differentiation. The topic is interesting and while the literature is rich about the roles of Polycomb in several system, this is an open question in the field. The experimental approaches are elegant even though sometimes, in my vision not solid enough to draw conclusions (see below). Thus, while the story is interesting and of worth there are missing points and limitations that at this stage dampen the enthusiasm for the manuscript in this form.

Authors' response:

We thank the reviewer for highlighting the interest in the topic and for recognizing the elegance of our experimental approach. We address the specific comments in the following. (Please note that this manuscript is by Hoffmann et al., not Matsuda and colleagues.)

Reviewer's comment:

Major points

The experiments regarding NSC differentiation in my vision are not solid as required to be one the key data of the work. Figure 3F-J: Differentiation in the control case (EhOH) should be equal in both lines (gray bars in H and J) but here it does not seem the case. In particular the *Pcgf3/5* line is less efficient. Considering the small phenotype that is present in *Pcgf2/4* line, is the poor differentiation of *Pcgf3/5* line maybe masking the phenotype?

Authors' response:

The NSC lines were derived from different mouse lines (*Pcgf2/4* and *Pcgff3/5*, respectively), hence each requiring appropriate controls. In designing our experiments, we have taken great care to compare only control and experimental lines that were derived during the same experiment from the same mouse lines, analyzing n=4-9 biological replicates. The differences of control *Pcgf2/4* and *Pcgff3/5* lines in their propensity to give rise to neurons and glia may be biological in nature, again underscoring the necessity of applying the appropriate control lines. Since all results underwent rigorous statistical testing, as indicated in figure legends, we believe that we have demonstrated the robustness of our findings.

Reviewer's comment:

I would suggest to perform additional experimental approaches to verify the findings. In utero electroporation of the Cre in different time windows to evaluate neuro- and gliogenesis? Differentiation in human setting?

The additional setting will be also useful to validate the findings of the genomic analysis in a more compelling situation.

Authors' response:

To obtain insights into the *in vivo* relevance of our data, we have administered tamoxifen to pregnant female mice at E12.5 (first dose) and E13.5 (second dose) to induce Cre expression from the *Nes::CreER^{T2}* allele. We first conducted control experiments to ensure that tamoxifen itself did not induce any brain phenotypes. However, even though we could induce efficient deletion of *Pcgfs* with two doses of 4 mg of tamoxifen, this led to a reduction in brain size at E17.5, in the absence of any floxed allele in *Nes::CreER^{T2}* control mice (see Reviewer Figure 1A). This is in line with previous reports in the literature (Forni et al., *Journal of Neuroscience* 2006), suggesting that tamoxifen, in combination with Cre expression, is toxic in NPCs during brain development. With reduced doses of tamoxifen (a single dose of 1 or 2 mg), we did not observe any effect on overall brain size (Reviewer Figure 1A). However, this lower dose of tamoxifen did not induce efficient deletion of *Pcgf* proteins (Reviewer Figure 1B). Therefore, we were unable to address the *in vivo* function of *Pcgf* proteins during neocortex development with our model. Given the unspecific effect of tamoxifen *in vivo*, we confirmed that such an effect is not observed *in vitro* (Reviewer Figure

1C), where we can achieve efficient deletion of Pcgf proteins (see Figure 2D, E).

Reviewer Figure 1. Tamoxifen-induced reduction in brain size in *Nes::CreERT2* embryos.

(A) The rostral-to-caudal length of embryonic mouse brains measured at E17.5 following administration of different concentrations of Tamoxifen (Tam) at E13.5 (1x) or E13.5 and E14.5 (2x). Note that the brain size is reduced in the absence of a floxed allele. (B) Immunoblots of protein lysates from embryonic brains at E17.5 after administration of 1 mg of Tamoxifen *in vivo* at E13.5, using anti-Pcgf5 and anti-Vinculin antibodies. (C) Quantification of the percentage of Huc/d-positive cells of total DAPI-positive cells following differentiation of NSCs isolated at E12.5 from *Nes::CreERT2* mice, treated with Tamoxifen (OHT) *in vitro* upon induction of differentiation.

Reviewer's comment:

Regarding the RNA-seq experiment: I would suggest to compare gene expression at the day 4 of differentiation with the starting point. In this way one can evaluate at the molecular level the neurogenic induction. Moreover, the temporal profile of expression will be evaluated on these important genes (either that should be upregulated or kept low during differentiation). Considering the dynamic role and activity of Polycomb complexes this may underline specific and direct PRC1 role during differentiation. GSEA analysis may return more complete picture than GO in this context. GSEA regarding "neuron differentiation" or "cortical development" should be presented.

Authors' response:

While we appreciate that temporal analysis of gene expression may give a more detailed view, we have analyzed gene expression at 6 DIV, which is 4 days after induction of deletion (see Figure 4A). Moving even earlier may not give sufficient time for the reduction of Pcgf protein levels and downstream effects on gene regulation. In the revised manuscript, we have extended the gene expression analysis to include GSEA (see new Expanded View Figure 3). This analysis revealed differences in the enrichment of gene sets associated with NPCs and brain/neocortex development (Florio et al, Science 2015), in particular genes with high CpG promoters previously associated with PRC2-mediated H3K27me3 (Mikkelsen et al, Nature 2007; Meissner et al, Nature 2008). This further corroborates our findings that Pcgf2/4 proteins contribute to the regulation of NPC differentiation.

Reviewer's comment:

Minor

Please use the correct international nomenclature for genes and proteins.

Authors' response:

In addition to the Pcgf names, we have now added *Mel-18* (for *Pcgf2*) and *Bmil* (for *Pcgf4*) to all figures and upon first mention in the text. We hope that will aid the recognition of the respective PRC1 components and link them better to previous literature, which uses both names.

Reviewer #2

Reviewer's comment:

The manuscript by Hoffman et al. reports *in vitro* observations about proliferation and differentiation abilities of primary neural progenitors (NPCs) from wild type and mutant mouse lines that can be made double deficient in Polycomb PCGF subunits, either *Pcgf2* and *Pcgf4* or *Pcgf3* and *Pcgf5*. The work tries to determine whether canonical and variant (non-canonical) Polycomb Repressive Complexes of class I (PRC1) regulate distinctly the developmental potential of NPCs. PCGF2 and PCGF4 form heterodimers with RING1A or RING1B in canonical PRC1 complexes, whereas PCGF3 and PCGF5 are present, together with the RING1 paralogs, in some of the variant PRC1 complexes. Mutant NPCs proliferative activity was reduced, compared to wild type cells, in both situations although to a large extent in double *Pcgf2/Pcgf4* mutants. Likewise, larger differentiation alterations were observed for *Pcgf2/Pcgf4* KO NPCs than for *Pcgf3/Pcgf5* KO cells. Finally, transcriptome analysis of *Pcgf2/Pcgf2* KO cells showed gene expression changes enriched within GO terms expected when dealing with Polycomb targets.

Although the nature of Polycomb complexes displayed by distinct cell states within given cell lineages is far from being well characterised, the fragmented evidence available support the hypothesis of the authors. I only have a reservation, also mentioned by the authors, that by not considering variant PRC1 complexes containing PCGF paralogs PCGF1 and PCGF6 is hard to extend the conclusions that suggest that canonical PRC1s are functionally more relevant to the developmental potential of NPCs than variant PRC1 complexes. In this regard, it would be beneficial to determine PCGF1 and PCGF6 protein expression in fetal brain sections so that indeed could be stated that PCGF3 and PCGF5-containing PRC1 complexes made most of the activity of variant PRC1s in developing brain.

Authors' response:

We thank the reviewer for the summary of our results that they find to complement the currently incomplete view of PRC1 function in the literature. We realize the importance of providing information on the expression of PCGF1 and PCGF6. Even though we were unable to obtain specific antibodies that work in immunohistochemistry of brain tissue sections, we now include gene expression data for human and mouse *PCGF1* and *PCGF6* in specific cell types of the developing neocortex (see new Expanded View Figure 1). *PCGF6* was not included in the transcriptome data set by Fietz et al, PNAS 2012. Therefore, we also included data from Florio et al, Science 2015. *PCGF1* and *PCGF6* showed no or low expression in the VZ, both in human and mouse. We noted in the discussion that “It remains possible that additional non-canonical complexes (PRC1.1/1.6) may functionally contribute to the regulation of NSC fate [...]”.

Reviewer's comment:

Another concern is about the *in vivo* relevance of the observations in tissue culture experiments. This is important because of the seemingly weak differences. I would imagine that this should be feasible given the availability of mouse lines combining the inducible alleles and a tissue-specific recombinase.

Authors' response:

We refer to the response to reviewer 1, who also raised a similar point, and to the Reviewer Figure 1 (see above). We agree that analyzing the *in vivo* role of *Pcgf* proteins would be highly complementary to our *in vitro* results. However, the toxicity of tamoxifen in combination with Cre expression in NPCs of the developing neocortex, which was previously described in the literature (Forni et al., Journal of Neuroscience 2006), prevented the *in vivo* analysis.

Reviewer's comment:

Regarding transcriptional alterations in mutant cells I don't see why *Pcgf3/Pcgf5*-mutant NPCs have been excluded from the analysis. Even if it little changes could be anticipated, based on available evidence, which nevertheless points at observable phenotypes.

Authors' response:

While we appreciate that analysis of gene expression in *Pcgf3/5* mutant NSCs may give a more comprehensive view, we have observed a stronger phenotype for *Pcgf2/4* deletion in both proliferation and differentiation and have therefore concentrated on the transcriptional regulation via this sub-complex.

Reviewer's comment:

It would also be interesting to determine global RING1A and RING1B protein levels in *Pcgf2/Pcgf4* KO NPCs. On the other hand, given the differential contribution of H2AK119Ub during neurogenesis (important at early stages, not so important at late stages) it would also be convenient to document it in the present set of experiments.

Authors' response:

This is an important point, which we have addressed in the new panels in Figure 2F–I. Analysis of global protein levels revealed a significant decrease of Ring1b levels upon *Pcgf2/4* knockout in NSCs without any changes in global H2AK119ub1 levels (Figure 2F, G). These results mirror previous findings in mESCs, where KO of *Pcgf2/4* was accompanied by reduced Ring1b levels, without affecting global H2AK119ub1 (Fursova et al, Mol Cell 2019). In contrast, deletion of *Pcgf3/5* in NSCs did not affect global levels of Ring1b, but resulted in reduced global H2AK119ub1 levels (Figure 2H, I). Again, this mirrors what has been observed for *Pcgf3/5* deletion in mESCs (Fursova et al, Mol Cell 2019). These results are consistent with previous studies showing weak ligase activity of canonical PRC1 both *in vitro* and *in vivo* (Gao et al, Mol Cell 2012; Blackledge et al, Cell 2014).

Reviewer's comment:

Other points

Figure 2I–K: there is a lack of correspondence between tissue culture images (I) and plots (J,K). Three of the experiments indicate little differences between OHT- and EtOH-treated cultures but the images seem to show otherwise.

Authors' response:

We have included a more representative image matching the quantitative differences in the cultures (see new Figure, now Figure 2M).

Reviewer's comment:

Figure 3: Why the proliferative defects of *Pcgf2/Pcgf4* KO cells are much milder than what one could expect from those suggested by Figure 2? Also, why EtOH-treated *Pcgf2/Pcgf4* and EtOH-treated *Pcgf3/Pcgf5* cultures (i.e. controls close to wild type) differ in percentages of Huc/d+ cells?. Differences are within the range seen between EtOH- and OHT-treated *Pcgf2/Pcgf4* cells which is interpreted as functionally relevant.

Authors' response:

We would like to point out that Figure 2 focuses on proliferation, whereas Figure 3 analyses the differentiation of NSCs. Thus, NSCs were cultured in different media, promoting proliferation or differentiation, respectively. Upon seeding in differentiation medium, NSCs typically only divide one more time before undergoing terminal differentiation. The knockout was induced concomitant with the seeding of NSCs in differentiation medium. Hence, the experimental setup related to Figure 3 does not allow the observation of proliferation phenotypes, and the presented images are a reflection of the plating of equal numbers of cells. Regarding the second point, as discussed in response to reviewer 1, the NSC lines were derived from different mouse lines (*Pcgf2/4* and *Pcgf3/5*, respectively), hence each requiring appropriate controls. We have, thus, made sure to select the correct respective controls.

Reviewer's comment:

Figure 4F: I am sure that NPCs derived from embryonic stem cells and primary NPCs derived from fetal brain are probably similar. However, it might have been preferable to look for chromatin occupancy data from primary NPCs so that panel F can be said truly relevant.

Authors' response:

As the reviewer points out, NPCs derived from ESCs and NPCs isolated from the developing brain are highly similar, for example, with respect to the transcription factors (Sox2, Pax6 etc) that they express. Thus, in lack of any more representative data sets, the PRC1 occupancy data in Figure 4F gives an indication of potential direct PRC1 target genes. To be transparent, we now specifically refer to “NPCs derived from embryonic stem cells (Figure 4F)”.

Reviewer's comment:

Figure 5: a more generous legend is necessary, in particular to the origin of data plotted in panels B-F.

Authors' response:

Thank you for pointing out this omission. We have included more specific information in the figure legend to enhance clarity.

Reviewer's comment:

Minor

Figure 1H lacks magnification bar.

Authors' response:

The scale bar has been added.

Reviewer's comment:

The paper by Fasano et al. 2007 has been retracted by the authors.

Authors' response:

We thank the reviewer for bringing the retraction of the Fasano et al. (2007) paper to our attention. We have removed this reference from our manuscript.

January 28, 2025

RE: Life Science Alliance Manuscript #LSA-2024-03006-TR

Dr. Mareike Albert
TU Dresden
Center for Regenerative Therapies Dresden (CRTD)
Fetscherstr. 105
Dresden 01307
Germany

Dear Dr. Albert,

Thank you for submitting your revised manuscript entitled "Canonical and non-canonical PRC1 differentially contribute to regulation of neural stem cell fate". We would be happy to publish your paper in Life Science Alliance pending final revisions necessary to meet our formatting guidelines.

- please be sure that the authorship listing and order is correct
- LSA allows supplementary figures, but no EV Figures; please update your callouts for the Supplementary Figures in the manuscript Fig EV1A=Fig S1A; while supplementary figures use the system supplementary Fig S1
- please note that the titles in the system and manuscript file must match
- please consult our manuscript preparation guidelines <https://www.life-science-alliance.org/manuscript-prep> and make sure your manuscript sections are in the correct order
- please add callouts for Figures S1A-D; SA-C and S3A-C to your main manuscript text-the RNA-seq data should be made publicly accessible at this point, removing the need for the Reviewer access token in the Materials and Methods

FIGURE CHECK:

- please add sizes to the blots in Figure 2
- blots in Figure S2 are actually source data, so please relabel and upload as such

LSA now encourages authors to provide a 30-60 second video where the study is briefly explained. We will use these videos on social media to promote the published paper and the presenting author (for examples, see <https://docs.google.com/document/d/1-UWCfbE4pGcDdcgzcmiuJI2XMBJnxKYeqRvLLrLS08s/edit?usp=sharing>). Corresponding or first-authors are welcome to submit the video. Please submit only one video per manuscript. The video can be emailed to contact@life-science-alliance.org

A. FINAL FILES:

B. MANUSCRIPT ORGANIZATION AND FORMATTING:

Thank you for your attention to these final processing requirements. Please revise and format the manuscript and upload materials within 5 days.

Sincerely,

January 31, 2025

RE: Life Science Alliance Manuscript #LSA-2024-03006-TRR

Dr. Mareike Albert
TU Dresden
Center for Regenerative Therapies Dresden (CRTD)
Fetscherstr. 105
Dresden 01307
Germany

Dear Dr. Albert,

Thank you for submitting your Research Article entitled "Canonical and non-canonical PRC1 differentially contribute to regulation of neural stem cell fate". It is a pleasure to let you know that your manuscript is now accepted for publication in Life Science Alliance. Congratulations on this interesting work.

DISTRIBUTION OF MATERIALS:

Again, congratulations on a very nice paper. I hope you found the review process to be constructive and are pleased with how the manuscript was handled editorially. We look forward to future exciting submissions from your lab.

Sincerely,
